# The Influence of Port Tugs on Improving the Navigational Safety of the Port

**Vytautas Paulauskas \*, Martynas Simutis, Birute Plačiene, Raimondas Barzdžiukas, Martynas Jonkus and Donatas Paulauskas**

Marine Engineering Department, Klaipeda University, H. Manto 84, LT-92219 Klaipeda, Lithuania; martynas.simutis@gmail.com (M.S.); birute.placiene@gmail.com (B.P.); logit2@inbox.lt (R.B.); m.jonkus123@gmail.com (M.J.); paulauskasd75@gmail.com (D.P.)
\* Correspondence: vytautas.paulauskas@ku.lt

**Abstract:** Port tugs are an important element in port activity and navigational safety issues. Port tugs ensure the safety of big ships while they are entering, manoeuvring, mooring and unmooring, and are of huge importance during other port operations. At the same time, optimizing the number of port tugs and tug bollard pull is also important from a port navigational safety and economic point of view. Calculation and evaluation methods of the optimal request for tugs bollard pull, in particular, port operations, are very important in order to guarantee the navigational safety of the port and ships during the main ship operations in the port. This article provides the number of requested port tugs and bollard pull calculation and evaluation methods on the basis of forces and moments acting on ships. On the basis of real ship voyages and manoeuvring at ports data as well as high accuracy simulators, theoretical methods were used, which were followed by our conclusions and recommendations, which can be used by port harbour masters and tug companies. Modern tugs have become an important element and integral part of modern port navigational safety. Such modern port tugs are also used for navigational safety and other important port functions and activities, such as fire protection and search and rescue operations. The optimal number and capacity evaluation of port tugs depending on port capacity and conditions are studied in this article.

**Keywords:** tugs; ship manoeuvring in ports; parametric optimization; ship mooring operations; optimal number of tugs and bollard pull; ship propulsion

## 1. Introduction

Safe operations are very important to the normal functionality of ports. Among the most difficult operations in ports are ships entering ports, mooring and unmooring operations, where the tugs are of the utmost importance. Port tugs assist ships using the port channels, manoeuvring of ships turning at basins, shifting to and from quay walls. Today, many ships have thrusters, which replace some of the tug functions, but many ships, especially tankers, bulkers and other big ships, do not possess such thrusters, which is why tugs are very important when it comes to the improvement of navigational safety [1–9].

Today, the tugs operated in ports are of a different type and capacity and mostly depend on ship size and port-external conditions (wind, waves, current and shallow water). The main risks at ports, which are pointed out by some authors, can be classified as follows: poor ship and port staff knowledge and training; the human factor in general; poor maintenance of port tugs; poor communication between all players during a ship's arrival at or departure from the port, as well as mooring operations (in the case that the ship's crew, port pilot and tugs masters communicate in different languages); poor or outdated tug equipment; poor safety culture, etc. [3,8,10–13]. According to Indian researchers [11], the risk factors and frequency at Indian ports are presented in percentages in Figure 1 below.

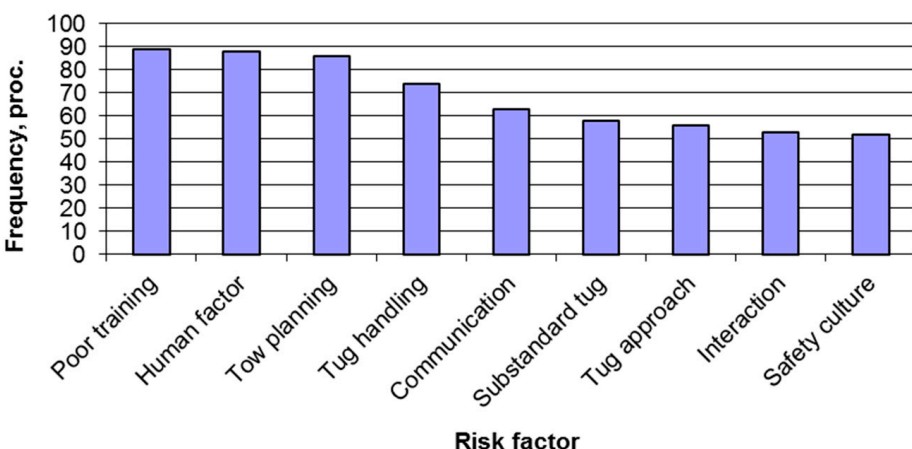

**Figure 1.** Risk factors and frequency at Indian ports in percentages.

Port tugs have an influence on risks factors in many cases, and this study aims to improve tug possibilities and decrease navigational risks in port areas. Such decrease in risks at ports is important issue to overcome, since the correct and proper usage of port tugs could highly improve the situation there.

Many ports today have a large number of sufficiently strong tugs (which depends on bollard pull). Tugs can be divided by their types as follows [9]: escort tugs—the field application of these tugs is generally to escort and help to manoeuvre barges or other vessels to their destination; support tugs—the field application of these tugs is to provide towing and berthing services to ports. Support tugs can be divided into three types (often, these types of tugs are named port tugs). 1. Conventional tugs—equipped with one or two propellers mainly used for push–pull assistance and towing operations, most suitable to assist small- or medium-sized vessels. 2. Azimuth stern drive tugs (ASD)—equipped with two thrusters; each can independently rotate at an angle of 360°. So, the thrusters can give thrust in every direction. They are very effective in open waters and during bollard pull operations. 3. Tractor tugs—equipped with two multidirectional propulsion units, and due to this feature, these are the only tugs which can provide full trust over 360°. Due to their good manoeuvrability and power ratio, these tugs are mainly used in port areas where they can easily handle the towage and berthing operations of megaships. For example, the table below provides the main characteristics of tug fleet at Klaipeda port [14] (Table 1).

**Table 1.** Tug use in Klaipeda port.

| Tug Name | Length (LOA) | Width | Draft | Engine Power | Bollard Pull |
|---|---|---|---|---|---|
| KLASCO1 | 29.44 m | 10.1 m | 4.6 m | 3370 kW | 550 kN |
| KLASCO2 | 29.44 m | 10.1 m | 4.6 m | 3370 kW | 550 kN |
| KLASCO3 | 27.9 m | 9.8 m | 4.6 m | 3728 kW | 600 kN |
| TAK4 | 26.39 m | 8.8 m | 4.8 m | 1297 kW | 300 kN |
| TAK5 | 24.22 m | 8.8 m | 4.8 m | 1297 kW | 350 kN |
| TAK6 | 26.61 m | 9.5 m | 4.8 m | 3430 kW | 550 kN |
| TAK10 | 28.67 m | 10.4 m | 4.6 m | 3728 kW | 610 kN |
| TAK11 | 28.64 m | 10.4 m | 4.7 m | 3728 kW | 610 kN |
| SL TENGIZ * | 48 m | 13 mm | 4.6 m | 2880 kW | 500 kN |

Remark * SL TENGIZ renders its services only to tankers, which are moored to the single mooring point (SPM).

Tugs in the ports are capable of generating forces greater or equal to the created forces (wind, waves and current and shallow water effect forces) on the vessel. In this case, it is very important to have instruments to calculate and evaluate forces (ship resistance, propulsion, etc.) that act on ships in particular conditions and requested tug capacities, which are able, together with the ship's propulsion system, to compensate forces acting on the ships under specific conditions. This paper presents methodologies for calculating



and estimating the forces and moments that are generated by ships in port conditions, as well as forces and moments generated by port tugs. A case study of the towage situation in complicated conditions confirmed the applied theoretical method. The obtained calculation and experimental data and their evaluation are necessary for the analysis of the methodology presented in the article and for the application of the methodology for practical purposes of the optimal use of port tugs. During ships' mooring and unmooring processes, it is necessary to accurately calculate the additional forces and capacities required to compensate the forces created by wind, waves, current and shallow water effects on ships.

At the same time, the lack of practical recommendations and methodologies for the optimal number of tugs and bollard pull calculations may result in unreasonable risks or excessive measures being taken in real life situations. The main objective of this article is to study and suggest practical methods as well as suggest optimal decisions on how to use tugs in ports and decrease the potential risks during ship manoeuvring operations in complicated conditions.

## 2. Literature Review

Port tugs are important for port navigational safety, and different approaches to estimate the quantity and quality of requested tugs in ports (bollard pull) have been implemented [5,7,8,15–17]. This also applies to transport and logistics systems functioning in pursuit of the sustainable development of these systems [4,15,18,19]. Direct and indirect tugs used in port areas are associated with navigational safety [1,9,20–22].

A number of studies address an interesting tugboat scheduling problem considering uncertainty in both container ship arrival and tugging process times for large container ports. For a large-scale problem, an ad hoc algorithm is designed to generate tugging chains such that the large-scale problem can be tackled effectively [5,23].

The aim of many papers is to rank the vessels entering and leaving the restricted channel of multiharbour basins and generate the optimal traffic scheduling schemes for each vessel, so as to ensure the safety and efficiency of vessel navigation. In these studies, through analysis of the characteristics of a restricted channel in ports, a general structure of a restricted channel in multiharbour basins is proposed, and the key areas of vessel traffic conflict are specified [13,24–28].

The papers describe the main ships' navigation processes and operations related to the port infrastructure and review the port simulation models [1,2,19,26,29–32]. This survey represents a detailed review of the state-of-the-art simulation models for port assessment purposes with a specific focus on safety and capacity. The model assessment focuses on the identification of the relevant criteria to represent a vessel's navigation based on which processes are covered by each model and how they have been considered in each model. The assessment provides an overview of the nautical infrastructure and navigational behaviour.

Generalized ship manoeuvring models based on the real-time modelling of ships navigating in ports are accessed and presented in [10,11,19,32–38]. Two models of the prediction of ships' trajectories have been developed and took into account the probability of ships leaving the channel or encountering navigational obstacles [19]: (1) an Auto Regressive and Moving Average eXogenous (ARMAX) model is adopted to identify the ship steering dynamic system; (2) the stochastic sequences of the inputs for the first model used are generated using a semi-Markov model. The papers describe the implementation of the semi-Markov model for rudder actions.

Manoeuvring models are dedicated for the rapid estimation of hydrodynamic factors in deep and shallow waters and allow a rapid estimation and reconstruction of the vessels' sailing trajectories for single and double propeller vessels [33,38–41]. Results are validated against experiments available for the zigzag and turning cycle trajectories of vessels with different hull forms and propulsion configurations [38].

Model predictive manoeuvring control and energy management for all-electric autonomous ships also aim to bridge the gap among manoeuvring control, energy management and the control of the Power and Propulsion System (PPS) in order to improve fuel efficiency and the performance of the vessel [33]. In this regard, for the ship motion control, a Model Predictive Control (MPC) algorithm is proposed which is based on Input–Output Feedback Linearization (IOFL). Through this algorithm, the required power for the ship mission is predicted and then transferred to the proposed Predictive Energy Management (PEM) algorithm, which decides on the optimal split between different on-board energy sources during the mission. As a result, the fuel efficiency and the power system stability can be increased.

Linear heave and surge movements recorded lower amplitudes compared to the values of standard thresholds [18]. The specific behaviour of each vessel was analysed in terms of its size, maritime conditions and mooring location. Field campaigns, such as those performed in this work, are an effective way of analysing the operational conditions of ports, which could help in identifying problems in the mooring zone [42–44].

The wave effects on ships moored in ports and a hybrid numerical model are proposed to estimate the transient response of a moored ship exposed to the two types of waves. The hybrid method is based on the combination of the 3D Rankine source method and impulse response theory. The 3D Rankine source method is applied to address the wash waves and the wave–structure interactions. The transient response is subsequently simulated in the time domain with the impulse response theory [45–47].

Human knowledge and experience accompanied by the ability to simulate the correct use of tugs in ports are of the utmost importance. These issues have been investigated by many researchers and seen by them as one of the key conditions for the correct use of tugs in ports [2,3,11,12,19,26,29,31,48,49].

It should be noted that port configuration and ship manoeuvring areas are different in particular ports [13,14,19,20,22]. The main factors influencing the optimal use of port tugs are as follows: types of manoeuvring operations performed by ships and the efficiency of tug assistance. On the one hand, ships moving in port areas have to be safe. Therefore, it is very important to optimize the time of ship movement and minimize manoeuvres inside the port that mainly depend on the qualifications of people in charge [3].

On the basis of the conducted literature analysis, the following statements can be made:

— The problem of decreasing or optimizing the use of port tugs is relevant, and further solutions in this area should be developed;
— There is a need to look for solutions to reduce (optimize) the use of port tugs that would not require high volumes of investments;
— To date, the influence of the human factor on port tug optimization has not been analysed to the required degree;
— The need to investigate the impact of a ship's crew and port pilots' qualifications and decisions on ship manoeuvring operations in port areas is justified, and further research in respect to how to decrease (optimize) the use of port tugs is required.

## 3. Theoretical Basis for Use of Port Tugs

Many ports in the world have and use port tugs. Port tugs could be classified depending on the capacity (bollard pull), type of propulsions (conventional or tractor type), functionality (just for tug operations or multipurpose) and ice class (no ice class or with ice class). Nowadays, it is common to build multipurpose tugs to avoid having additional ships with very narrow functionalities, such as fire protection ships and environmental protection vessels. Consequently, a substantial number of the aforementioned functions are entrusted to port tugs.

Port tug bollard pull ranges are from 50 kN (in small ports) to 700–1000 kN in ports, which accept mega vessels, such as E or G class container vessels, SUEZMAX or bigger tankers, capsize or bigger bulkers and other ships. Today, many ports use tugs, which have bollard pull of about 500–600 kN. For example, Klaipeda port, which has an annual

turnover of about 48 million tons and accepts various types of ships, such as large E and G class container ships, SUEZMAX tankers and bulk cargo POST PANAMAX ships, has eight port tugs with a traction force (bollard pull) of up to 650 kN [14].

Tug-related risks, according to many researchers and experts, could be classified into risk factors, risk factor frequency and the relationship between risk factors and accidents during towage operations in ports [6,16]. Crew incompetence due to poor crew preparation is observed in more than 75–80% of accidents and incidents in towage operations, and it could be considered as a strong negative factor [3,12,37,45]. In many countries, tug masters and other tug personnel have very limited knowledge about the manoeuvrability of big sea-going ships. The lack of towage process planning in advance is a very important factor and accounts for up to 80% ship accidents and incidents due to mistakes made in the preparation of towage-related processes [3,11,45,49]. A lack of practice on the part of tug personnel for sea-going ships and insufficient knowledge about the manoeuvrability of big sea-going ships are observed in more than 40% accidents and incidents occurring during towage operations [6,16]. It should also be noted that accidents and incidents during the towage process that took place due to weather conditions and poor visibility, also mentioned as root causes, account for about 20% [11,12,45].

When mooring vessels at berths, it is necessary to accurately calculate the additional forces and capacities required to compensate for external forces.

For ships moored to the quay and from them, it is necessary to calculate the necessary additional power and output of the compensated external impact forces.

Generally, the following forces act during ship manoeuvring in port areas and mooring operations: the forces of inertia of the ship (when stopping the ship or giving it the required speed); the forces of direct action of the hydrodynamic current; hydrodynamic "wing" forces (when the vessel is moving upstream, or standing in the current); aerodynamic forces; shallow water effect forces, etc.

Forces and moments created by wind, current, waves and shallow water effects acting on the ship are compensated by generating ship propulsion system forces and moments with the help of the tugs being used. Thus, the calculation of the forces and moments can be performed using the following mathematical model [6,7,50]:

$$X_{in} + X_k + X_\beta + X_p + X_N + X_a + X_c + X_b + X_{sh} + X_T + X_{tug} + \ldots = 0 \tag{1}$$

$$Y_{in} + Y_k + Y_\beta + Y_p + Y_N + Y_a + Y_c + Y_b + Y_{sh} + Y_T + Y_{tug} + \ldots = 0 \tag{2}$$

$$M_{in} + M_k + M_\beta + M_p + M_N + M_a + M_c + M_b + M_{sh} + M_T + M_{tug} + \ldots = 0 \tag{3}$$

where $X_{in}, Y_{in}, M_{in}$—inertia forces and the moment; forces and moment created by the ship's hull $(X_k, Y_k, M_k)$ could be calculated by using the methodology stated in [51–53]; $X_\beta, Y_\beta, M_\beta$—the ship's hull as the acting "wing"-related forces and the moment could be calculated using the methodology stated in [50]; $X_p, Y_p, M_p$—forces and the moment created by the ship's rudder or other steering equipment; $X_N, Y_N, M_N$—forces and the moment created by thrusters; $X_a, Y_a, M_a$—aerodynamic forces and the moment could be calculated using the methodology stated in [52,53]; $X_c, X_c, M_c$—forces and the moment created by the current could be calculated using the methodology stated in [6,26]; $X_b, Y_b, M_b$—forces and the moment created by waves could be calculated using the methodology stated in [51–53]; $X_{sh}, Y_{sh}, M_{sh}$—forces and the moment created by shallow water effects; $X_T, Y_T, M_T$—forces and the moment created by ship's propeller (propellers) could be calculated using the methodology stated in [51–53]; $X_{tug}, Y_{tug}, M_{tug}$—forces and moment created by tugs. Additional forces and moments could be created by anchor or mooring ropes or other factors.

Tug and ship steering equipment creates forces and moments, which must compensate other forces and moments acting on the ship. The system of Equations (1)–(3) could be presented as follows:

$$X_{in} + X_k + X_\beta + X_a + X_c + X_b + X_{sh} + X_T + \ldots = X_p + X_N + X_{tug} \tag{4}$$

$$Y_{in} + Y_k + Y_\beta + Y_a + Y_c + Y_b + Y_{sh} + Y_T + \ldots = Y_p + Y_N + Y_{tug} \tag{5}$$

$$M_{in} + M_k + M_\beta + M_a + M_c + M_b + M_{sh} + M_T + \ldots = M_p + M_N + M_{tug} \tag{6}$$

Inertia forces and the moment $X_{in}, Y_{in}, M_{in}$ could be calculated using the methodology stated in [50]:

$$X_{in} = (m + \lambda_{11})\frac{dv_X}{dt}; \tag{7}$$

$$Y_{in} = (m + \lambda_{22})\frac{dv_Y}{dt}; \tag{8}$$

$$M_{in} = I_Z(1 + k_{66})\frac{d\omega}{dt} + \rho V(k_{11} - k_{22})v^2\beta, \tag{9}$$

where $m$—ship's mass; $\lambda_{11}$—add water mass in X direction; $v_X$—ship's speed in X direction; $\lambda_{22}$—add water mass in Y direction; $v_Y$—ship's speed in Y direction; $I_Z$—the moment of inertia depends on Z axe; $k_{66}$—inertia moment coefficient; $\omega$—angular rate of rotation of the vessel; $\rho$—water density; $V$— displacement of the ship in m$^3$; $k_{11}$—add water mass in X direction coefficient; $k_{22}$—add water mass in Y direction coefficient; $v$—ship's speed module; $\beta$—ship's drift angle.

Forces and moment created by the rudder (rudders) $(X_p, Y_p, M_p)$, acting on a ship's steering, could be calculated as follows [51,52]:

$$X_p = C_x\frac{\rho}{2}S_p v_s^2; \tag{10}$$

$$Y_p = C_y\frac{\rho}{2}S_p v_s^2; \tag{11}$$

$$M_p = Y_p l_p, \tag{12}$$

where $C_x, C_y$—rudder hydrodynamic coefficients; $\rho$—water density; $S_p$—the area of projection of the rudder plane into the diametrical plane; $l_p$—rudder transverse force shoulder.

The forces and moment created by shallow water effects $(X_{sh}, Y_{sh}, M_{sh})$ could be found by taking both a theoretical approach and carrying out a real ship investigation. At shallow depths, the manoeuvring characteristics of the vessel change due to the increased shallow effect, i.e., increased ship draft and resistance; the added mass of liquid (water) "touches" the bottom of the channel or water area at the same time increases the frictional resistance. The change in the ship's stopping performance, for example, can be expressed as a coefficient of resistance, which depends on the ship's draft-to-depth ratio $(T/H)$. When the ship is moving straight, its coefficient of resistance—due to the effect of shallowness—is stated in Figure 2 [50]. The experiments were performed on the real ships while comparing the speed of the ship at the same power used by the ship's engine, sailing at high $(H/T \geq 6)$ and shallow $(T/H \leq 6)$ depths. Vessel speed measurements were performed with the navigation equipment of high-precision E-Sea Fix. The tolerance applied did not exceed 0.1 knot or 1% of the initial speed of the vessel.

The ship's coefficient of resistance when the ship is moving straight $(k_{R11})$, taking into account $T/H$, can be calculated according to the following regression formula:

$$k_{R11} = 1 + 3.45\left(\frac{T}{H}\right)^2; R^2 = 0.91. \tag{13}$$

With the ship moving sideways, the increase in the coefficient of resistance, taking into account the ratio $T/H$ obtained with the help of experiments using real ships, is shown in Figure 3. During the experiments, the ship's drift angle was measured at high $(T/H \geq 6)$ and shallow $(T/H \leq 6)$ depths. Ship drift angle measurements were performed with the navigation equipment of high-precision E-Sea Fix. The tolerance applied did not exceed 0.1° or 1% of the ship's drift angle at high and shallow depths.

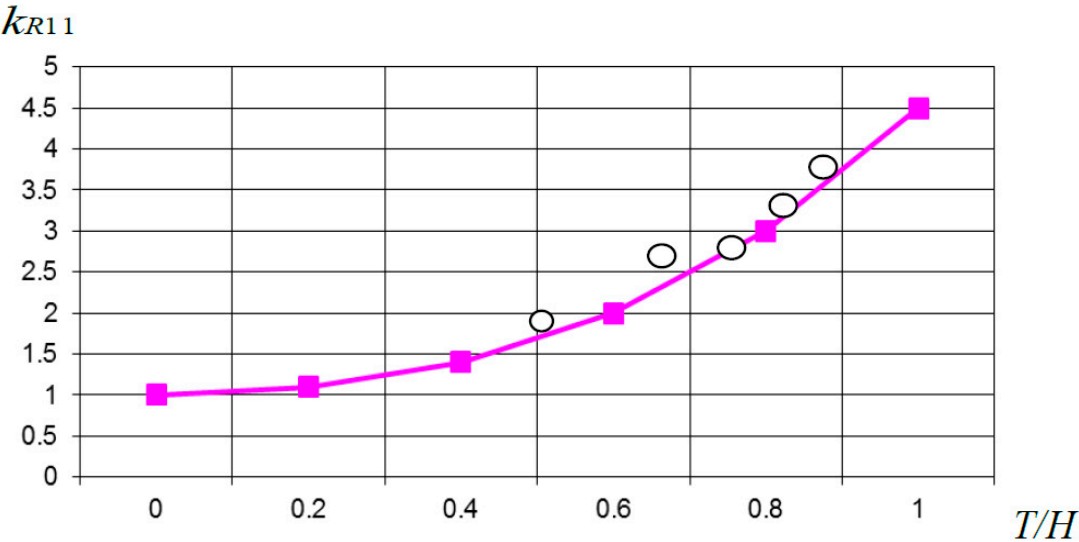

**Figure 2.** Ship's coefficient of resistance $k_{R11}$ depending on ship's draft and depth $T/H$ (calculation and experimental "o" results).

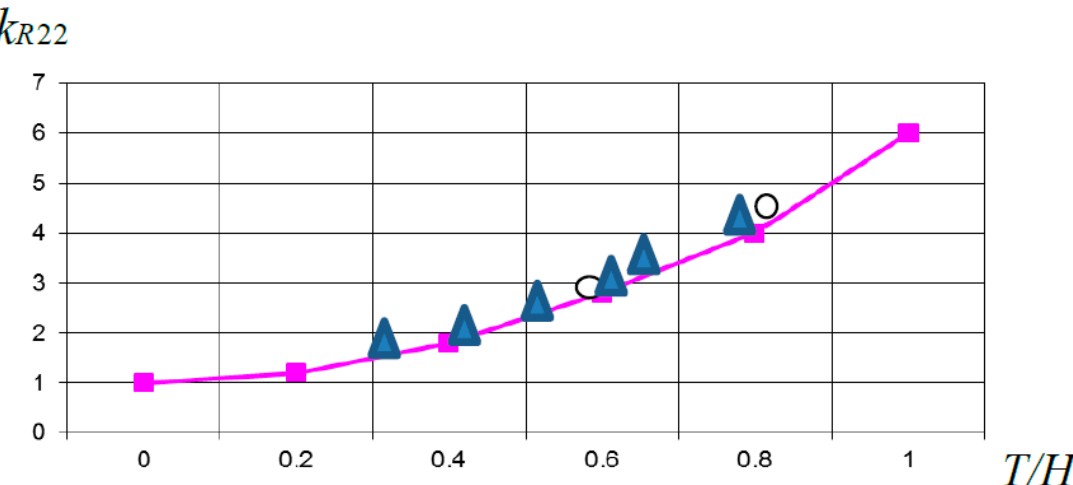

**Figure 3.** Ship's additional coefficient of resistance $k_{R22}$, depending on ship's draft and depth $T/H$ (calculation and experimental "o" for the tanker L = 250 m and ("▲") for the container vessel L = 365 m results).

The coefficient of resistance of a ship moving sideways ($k_{R22}$), taking into account $T/H$, can be calculated according to the following regression formula:

$$k_{R22} = 1 + 4.95 \left(\frac{T}{H}\right)^2; \ R^2 = 0.93 \tag{14}$$

The obtained formula of the ship's coefficient of resistance when the ship is moving sideways allows the possibility to calculate the real ship's resistance when mooring at or off the berth. This situation is particularly important when vessels are moored at shallow depths at piers located at a large angle (50° to 90°) in the direction of the current and in other situations. The additional lateral resistance of the vessel when moving the vessel is important in the study of channels at shallow depths, as this resistance reduces the drift speed of the vessel, ship turning, etc.

Studies of Formulas (13) and (14) obtained by the regression method revealed that the errors of the searched values using regression formulas in comparison with the results of experiments performed on real ships [19,40] do not exceed 5–7%.

The forces and moment created by thrusters $(X_N, Y_N, M_N)$ could be calculated as follows:

$$X_N = k_{NX} F_{NX}\left(1 - \frac{v'}{v'_0}\right) \tag{15}$$

$$Y_N = k_{NY} F_{NY}\left(1 - \frac{v'}{v'_0}\right) \tag{16}$$

$$M_N = Y_N l_N \tag{17}$$

where $k_{NX}$—thrusters coefficient in X direction, which could range from 0.01 to 0.1 depending on the thruster's installation position on ship; $F_{NX}$—thruster's force in X direction can be taken from thruster's specification; $v'_0$—ship effectiveness depending on thruster's speed, in many cases falling within 3–4 m/s; $v'$—ship's speed, which could range from 0 to $v'_0$; $k_{NY}$—thruster's coefficient in Y direction, which could range from 0.8 to 1.0 depending of the thruster's installation position on the ship; $F_{NY}$—thruster's force in Y direction can be taken from thruster's specification; $l_N$—thruster (s) transverse force shoulder.

Forces and moments created by the tug (tugs) $(X_{tug}, Y_{tug}, M_{tug})$, depending on the towage system (towage or pull/push), could be calculated on the basis of the fixed points of towage ropes or tugs on the ship. In the case of towage, forces and moments created by tugs (for each tug) could be calculated as follows:

$$X_{tug} = F_{tug} \cdot \cos\alpha \cdot \cos\beta; \tag{18}$$

$$Y_{tug} = F_{tug} \cdot \sin\alpha \cdot \sin\beta; \tag{19}$$

$$M_{tug} = Y_{tug} \cdot l_{tug}, \tag{20}$$

where $F_{tug}$—tug's bollard pull; $\alpha$—horizontal angle between ship's diametric plane and towage rope; $\beta$—vertical angle between tug's and ship's fixed points; $l_{tug}$—towing force shoulder.

The towing force shoulder can be found using the abscissa of the ship's turning pivot point ($x_0$) (Figures 4–6) and towage rope length ($l_{tr}$). The pivot point of the ship depends on the ship's movement direction, which means that the ship is moving ahead or astern and can be calculated as follows [16,50]:

$$x_0 = L\left(0.4 + 11.5\frac{T_{lg} - T_{lp}}{L} - 0.004 \cdot \alpha_r^0\right), \tag{21}$$

where $L$—ship's length between perpendiculars; $T_{lp}$—ship's bow draft; $T_{lg}$—ship's aft draft; $\alpha_r^0$—rudder turn angle in degrees.

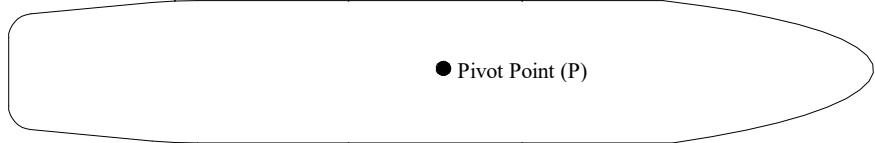

**Figure 4.** The location of the pivot point of the ship in the stop position.

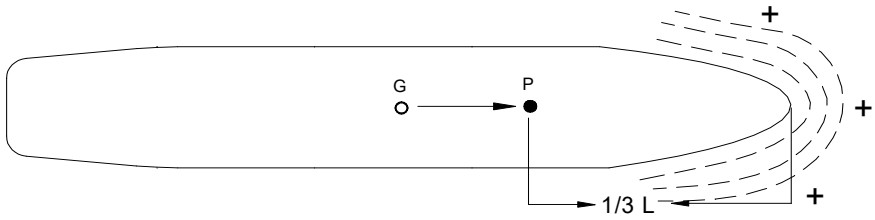

**Figure 5.** The location of the pivot point when the ship moves ahead.

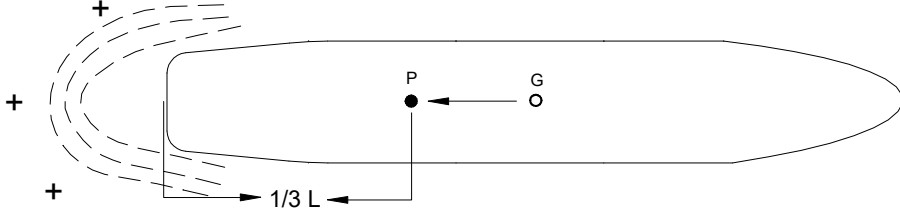

**Figure 6.** The location of the pivot point when the ship moves astern.

Finally, towing force shoulder can be calculated as follows:

$$l_{tug} = ((x_0 + \frac{L}{2}) + l_{tr} \cdot \cos\beta) \cdot \sin\alpha. \tag{22}$$

Experiments were performed with the help of a simulator and in real ships. The obtained data are necessary for the evaluation of the methodology and its practical application in calculating the required number of port tugs and their bollard pull. The studies used a three-step system, i.e., the calculation method presented in this part of the article, the simulations using the SimFlex Navigator simulator and the results of experiments performed in real ships under similar conditions. The maximum distribution method [6,29] was used to evaluate the accuracy and reliability of the obtained calculation and experimental results.

On the basis of the presented methodology, theoretical calculations of possibilities of the ship's entry and departure, manoeuvring, mooring and unmooring in complicated conditions and the minimum request for tugs (bollard pull) can be made. For checking theoretical calculations, a full mission simulator "SimFlex Navigator" (Force Technology product) was used, which also analysed similar manoeuvres of the real ships.

## 4. Case Study Tugs Used in Complicated Navigational Conditions at Klaipeda Port

For the case study, we opted for the unmooring and departure of the largest possible ship from berth No. 127 of Klaipeda port under different meteorological conditions.

According to Harbour Master's order, big ships can arrive at or depart from Klaipeda port, with wind limitation up to 14 m/s (30 s period wind speed) [14]. For the case study, the worst wind directions for this very port were taken: from the north wind direction (N), via the west wind direction (W) up to the south wind direction (S). In order to guarantee the navigational safety, the minimum possible tugs and bollard pull for the aforementioned wind speed and directions were calculated. For the case study, theoretically required bollard pull and the number of tugs were calculated, simulation tests were performed and similar real cases (with real ships) were verified.

To check the accuracy and reliability of the obtained results of the amount of tugs and the tensile force (bollard pull), a SimFlex Navigator simulator was used, and experiments were performed in vessels and tugs of similar parameters under analogous conditions performing the same ship mooring and sailing operations. During the experiments, the tug rope tension was measured in the tugs every 10 s with the help of the rope tension recorder. Using the push–pull method, the engine power of the tugs was recorded by the main engine recorder every 10 s. Based on the instantaneous power of the main engine of the tugs, the propulsive force of the ship's propeller (pushing–pulling force of the tug) was calculated. The obtained experimental results, after processing them by the method of maximum distribution, were compared with the calculation method and the results obtained by the simulator.

Quay wall No. 127 (jetty) (Figure 7) has a length of 265 m, with depths near the jetty reaching up to 10 m and depths in the bay −14 m. A container vessel of the following characteristics was used for the case study: length between perpendiculars: 238 m; beam: 32.2 m; draft: 9.2 m; displacement: 64,500 t; rudder plane into the diametrical plane: 36 m$^2$; ship's air projection on diametric square: 7200 m$^2$; ship's air projection on middle square: 950 m$^2$; the space of projection onto a diametrical plane of the underwater area of the ship:

2190 m$^2$; ship's bow draft: 9.1 m; ship's aft draft: 9.3 m; water density: 1000 kg/m$^3$. To assist the ship's unmooring and manoeuvring, from 1 to 3 tractor type tugs were used; each of the tugs had a bollard pull up to 500 kN (length of the tugs—29.44 m, width—10.1 m, draft—4.6 m, engine power—3370 kW). Tugs used the towage and pull/push methods depending on the area conditions. The tension of the tow rope was fixed by the tug's recorder; the pushing–towing forces were converted according to the actual engine power, which was fixed by the tug engine recorder. Next to the other berths, whilst leaving the manoeuvring area (bay), the POST PANAMAX cargo ship (bulker) and ferry were moored at the nearest berth (Figure 8).

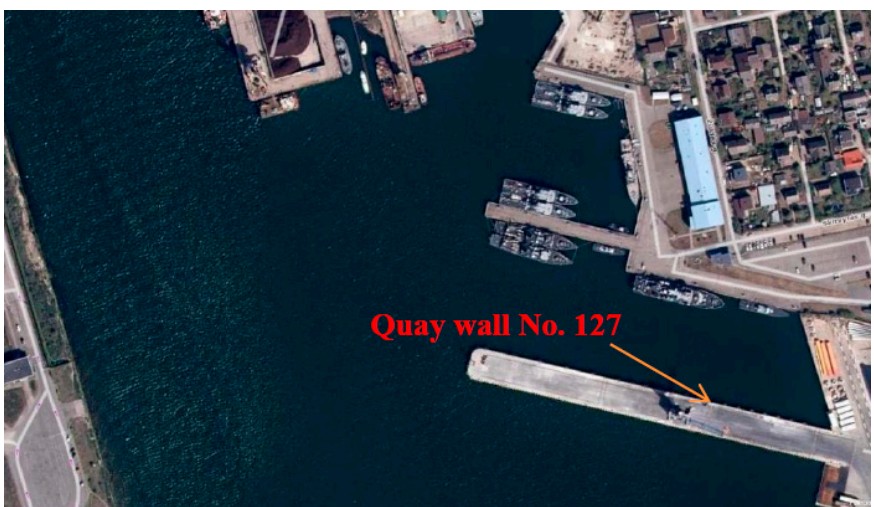

**Figure 7.** Navigational conditions at quay wall No. 127 area.

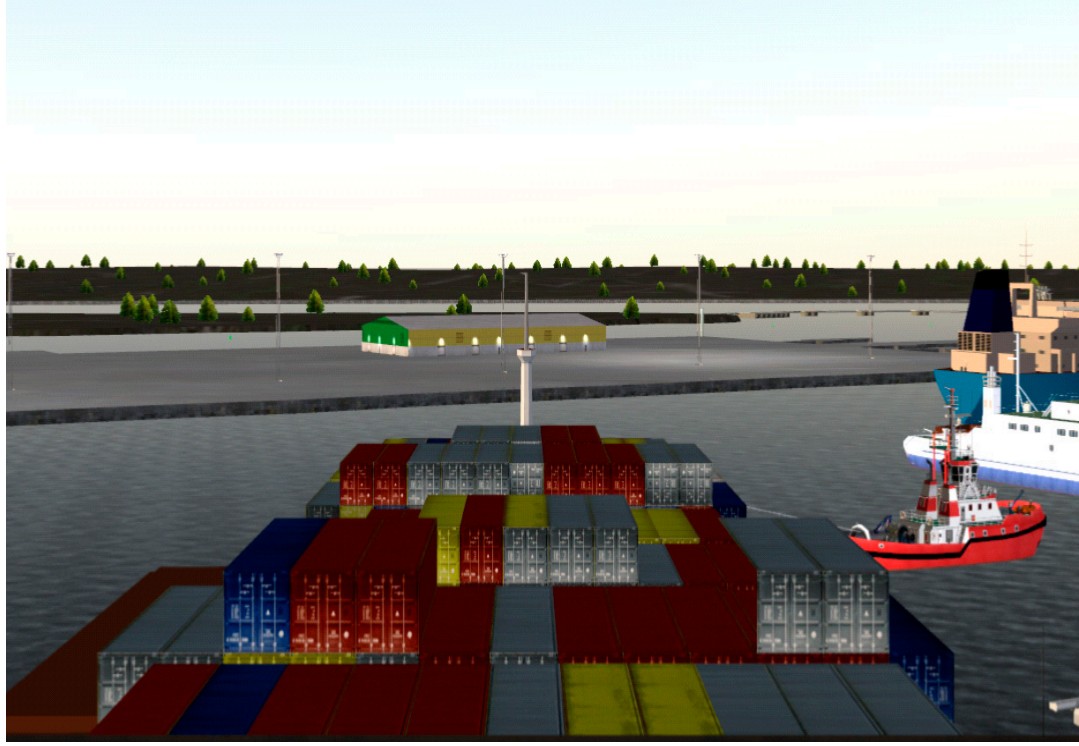

**Figure 8.** Visual situation of the area before unmooring operation.

For calculation purposes, characteristics of the quay wall equipment (fenders) and local requirements were taken into account [54,55]. The following conditions were chosen— no waves, no current, thrusters not used and ship having a very low speed. In view of the aforementioned characteristics and conditions used, Equations (4)–(6) can be adjusted and presented as follows:

$$X_{in} + X_k + X_\beta + X_a + X_{sh} + X_T + \ldots = X_p + X_{tug} \tag{23}$$

$$Y_{in} + Y_k + Y_\beta + Y_a + Y_{sh} + Y_T + \ldots = Y_p + Y_{tug} \tag{24}$$

$$M_{in} + M_k + M_\beta + M_a + M_{sh} + M_T + \ldots = M_p + M_{tug} \tag{25}$$

In this case study, the main factor is wind. For the selected ship (L = 238 m), the forces created by the wind (which were calculated using the methodology presented in this article) depend on the wind direction and speed and are presented in Figure 9.

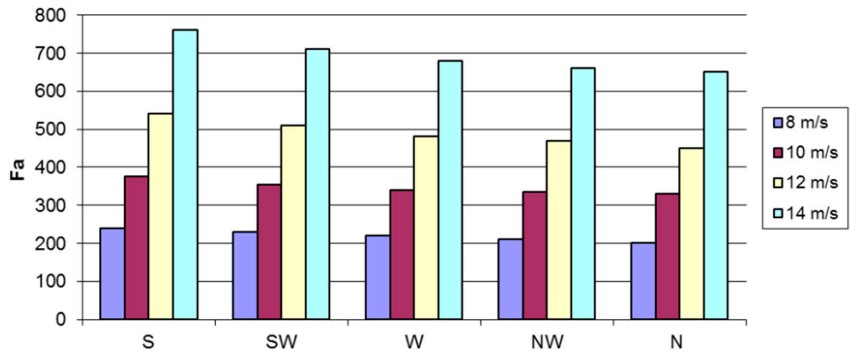

**Figure 9.** Wind forces on ship (L = 238 m) depend on wind direction and speed.

In the case of the north direction wind with a speed of 14 m/s, for the ship's unmooring and departure operations, the calculations made using the methodology presented in this article showed that it is enough to use one tug with bollard pull up to 500 kN and the ship's propulsion facilities. Figures 10 and 11, which show the calculation method, simulation and fixed tug bollard pull in real operations with a similar ship, present the ship's trajectory and tug bollard pull during unmooring, turning and passing from the bay via channel operations.

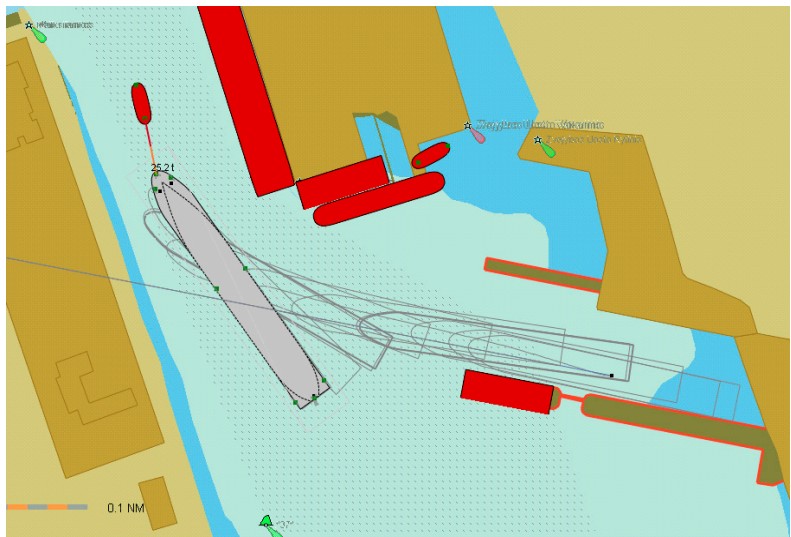

**Figure 10.** Ship's trajectory on departure using one tug with bollard pull up to 500 kN (wind N—14 m/s).

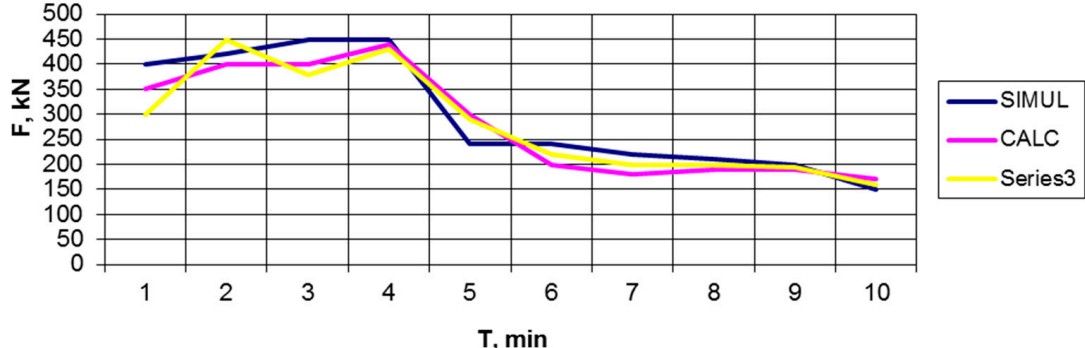

**Figure 11.** Tug's towage forces during ship's unmooring and departure operations, obtained by calculation (CALC) and simulation (SIMUL) during real operation with similar vessel under hydrometeorological conditions (Series3) (wind N—14 m/s).

In similar real ship operation conditions, 2 tugs were used, but at the same time, the maximum bollard pull or towage forces used by both tugs were not more than 450 kN (tugs used from 20 to 50% of the bollard pull during all unmooring and manoeuvring operations).

In the case of the west wind direction with a speed of 14 m/s, calculations made revealed that at least 750 kN bollard pull or towage forces during the ship's turning and keeping in the channel are required, because the ship must turn around and move astern so that it may subsequently gain sufficient speed for navigation via the channel from the bay.

Figures 12–14 present the ship's trajectory, whereas Figure 15 provides tugs towage forces during the ship's unmooring, turning and passing from the bay via the channel.

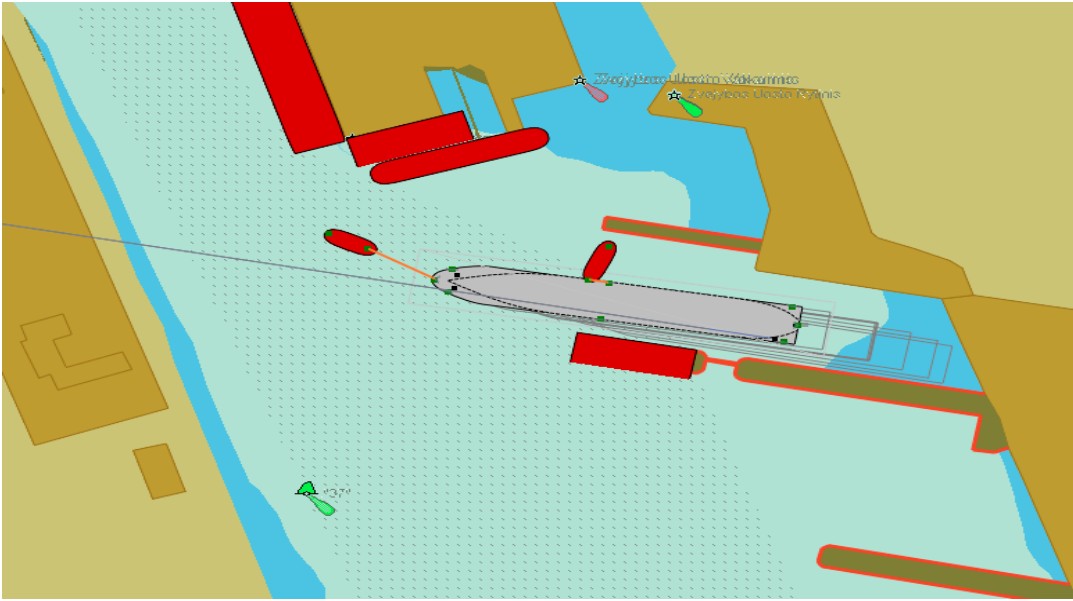

**Figure 12.** Ship (L = 238 m) departure from berth in the case of wind W—14 m/s (two tugs were used: the first for towage, and the second using the push/pull method).

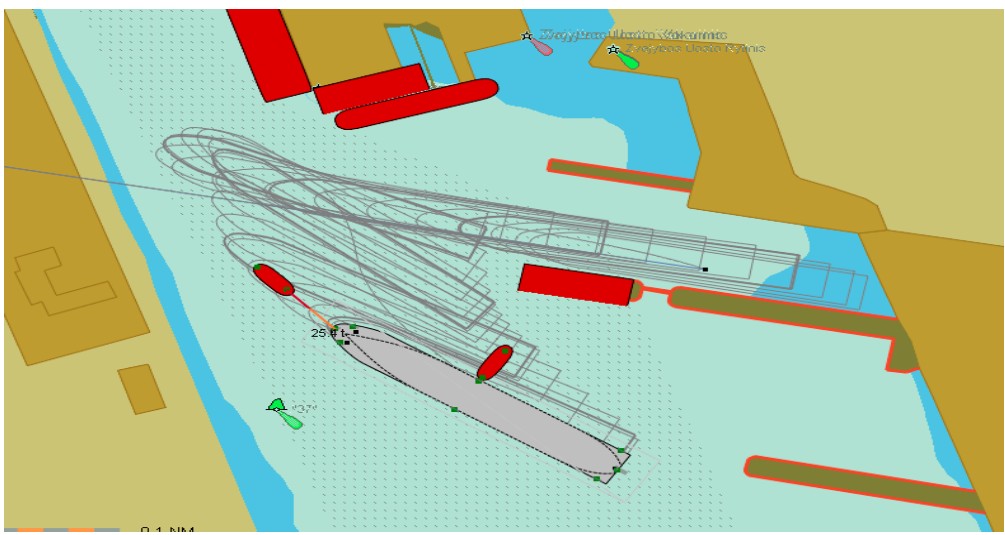

**Figure 13.** Ship's departure trajectory with wind W—14 m/s until tug No 2 stopped push/pull operation.

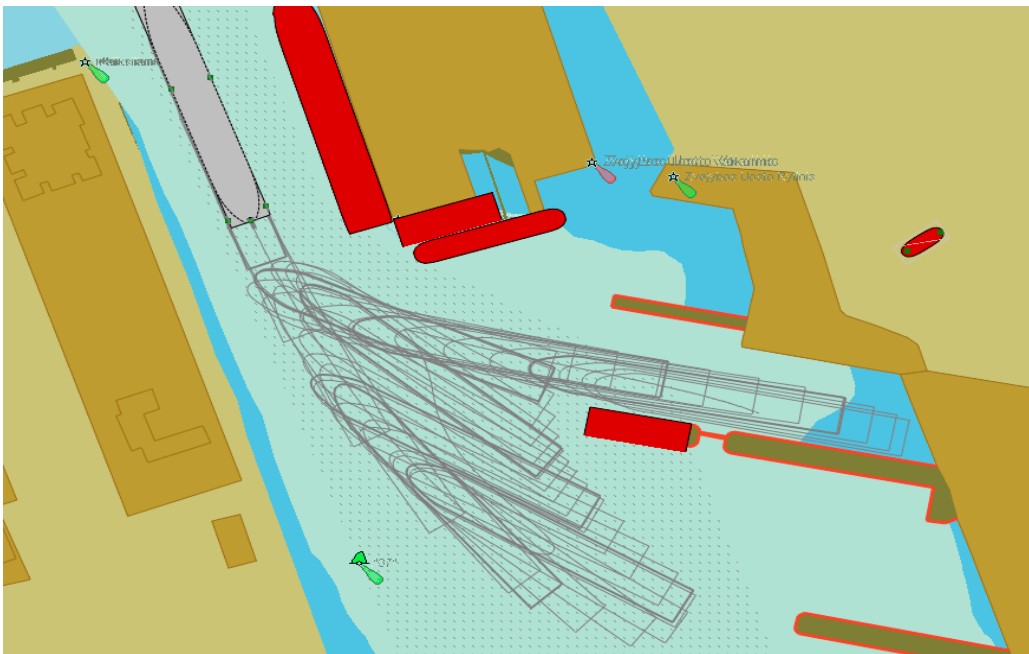

**Figure 14.** Ship's (L = 238 m) departure trajectory in the case of wind W—14 m/s.

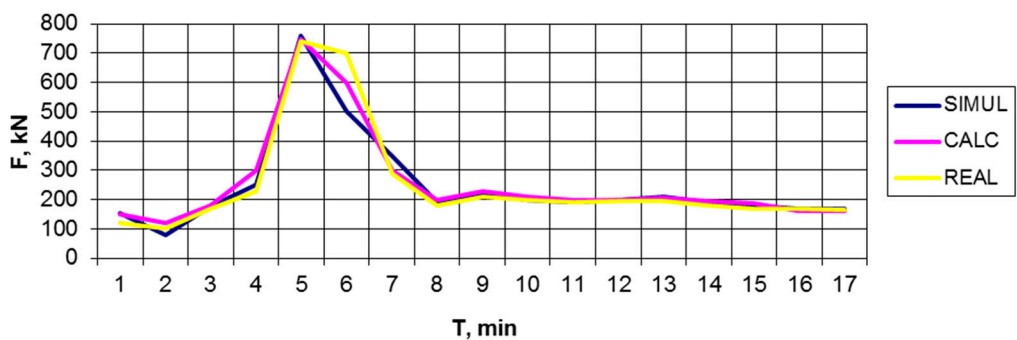

**Figure 15.** Tug towage line tension and pull/push forces during ship's unmooring, manoeuvring and departure operations, obtained by calculation (CALC) and simulation (SIMUL) during real ship operation with similar vessel under hydro meteorological conditions (REAL) (wind W—14 m/s).

In similar real ship operation conditions, 3 tugs were used, but at the same time, the maximum bollard pull used by all 3 tugs together did not exceed 700–750 kN (tugs used from 20 to 75% of the bollard pull during all unmooring and manoeuvring operations). It appears that the west direction wind is more complicated in comparison with the north direction wind.

In the case of the south direction wind, unmooring is more complicated, because the area between the moored ship and quay wall No. 125 is very small, and there is no possibility of a tug on the ship's stern for pushing the ship against its drift in the north direction. Calculations of the required bollard pull made in accordance with the methodology presented in this article reveal that 3 tugs and bollard pull in some instances reaching up to 1300 kN are required. At the same time, during the first stage of unmooring operations, there is no possibility of using three tugs, and only 1 tug can be used on the bow of the vessel for towage using a towage rope and 1 tug at the middle of the ship—for the push/pull operation. Figure 16 presents the tug location during the first stage of unmooring operations.

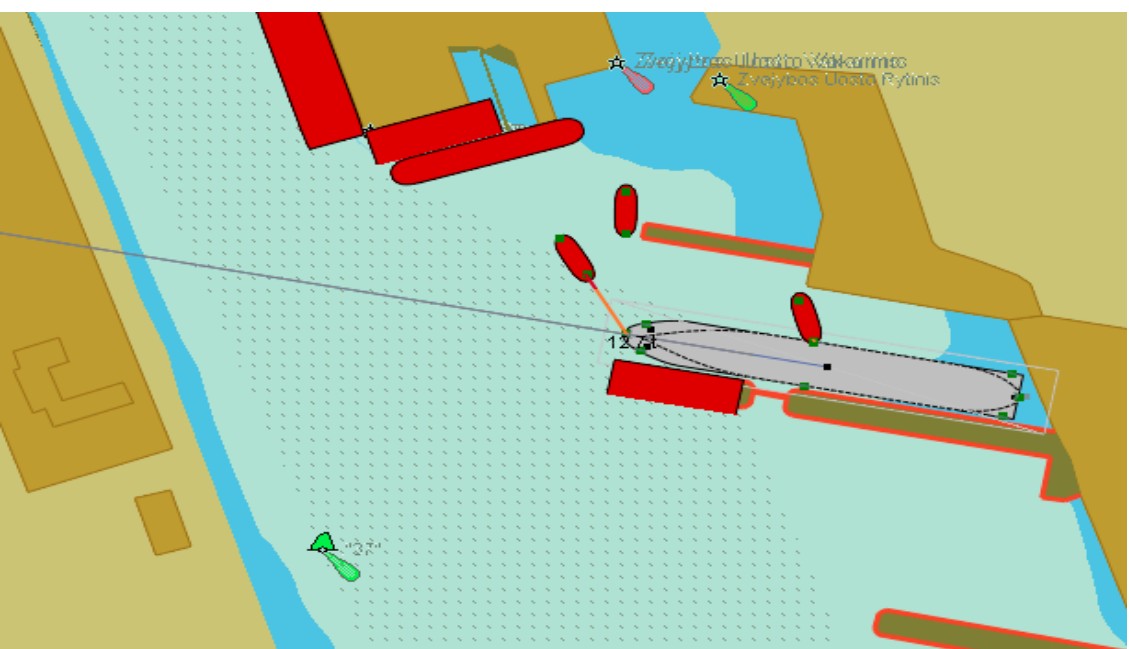

**Figure 16.** Ship (L = 238 m) departure from berth in the case of south direction wind—14 m/s and tugs location (three tugs were used: the first in towage position, the second in push/pull position and the third in stand-by position).

After the start of the unmooring operations and disconnection of all mooring ropes, the ship's position is shown in Figure 17. The acting bow tug, tug at the middle of the ship (push ship operation against her drift) and additional tug (tug No. 3) must be ready to start the push/pull operation when the ship's stern is close to the berth. In Figures 18–20 present the ship's trajectory, whereas Figure 21 provides tugs towage forces during the ship's unmooring, turning and passing from the bay via the channel.

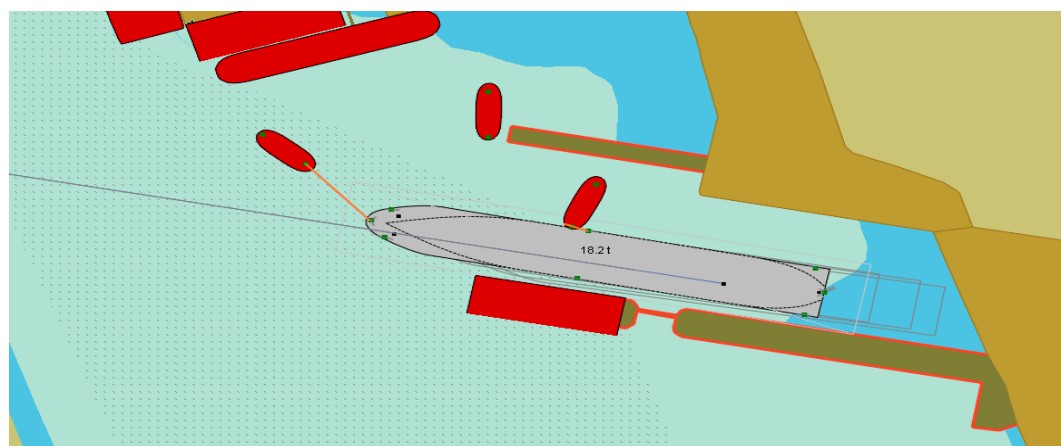

**Figure 17.** Ship (L = 238 m) moving off the berth with south direction wind of 14 m/s, and three tugs were used, located as follows: the first in towage position, the second in push/pull position and the third in stand-by position.

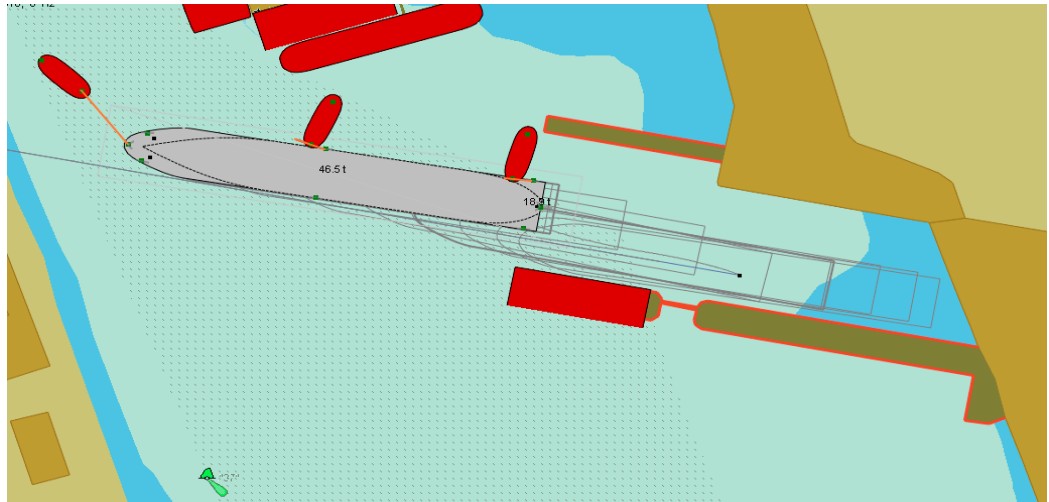

**Figure 18.** Ship (L = 238 m) moving off the berth with south direction wind of 14 m/s, and three tugs were used located as follows: the first in towage position and the second and third in push/pull position, all three tugs acting.

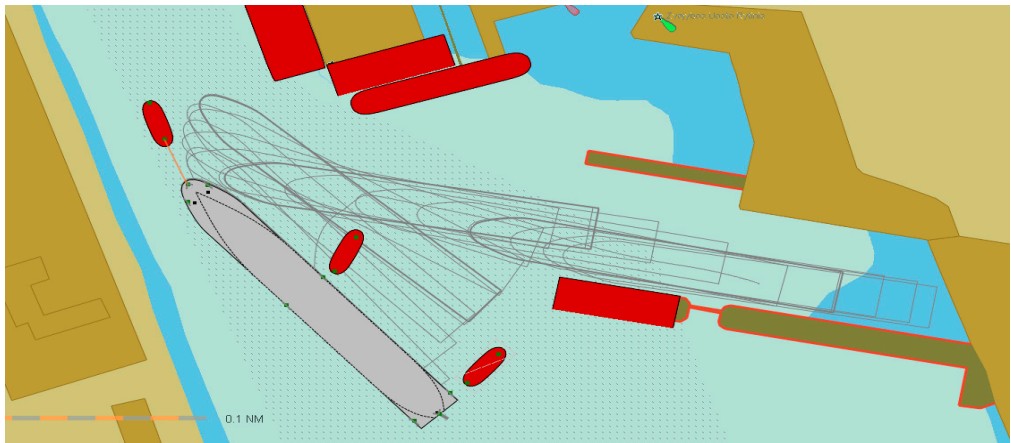

**Figure 19.** Ship (L = 238 m) moving off the berth with south direction wind of 14 m/s, and three tugs were used located as follows: the first in towage position, and the second and third in push/pull position and ending with push/pull operation.

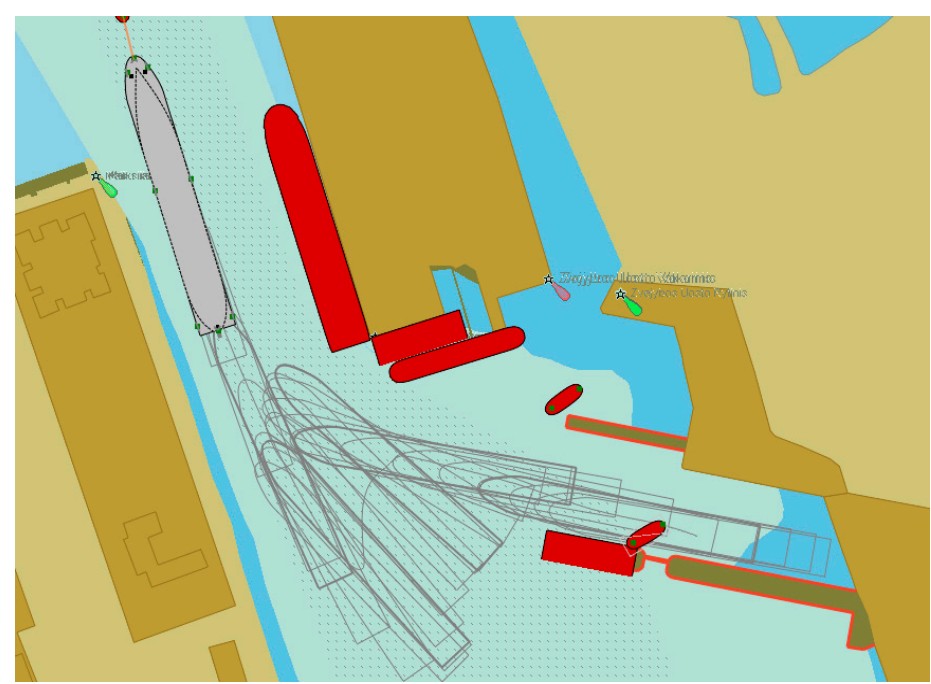

**Figure 20.** Ship (L = 238 m) moving off the berth with south direction wind of 14 m/s; trajectory of the vessel at the channel from bay with just one tug acting.

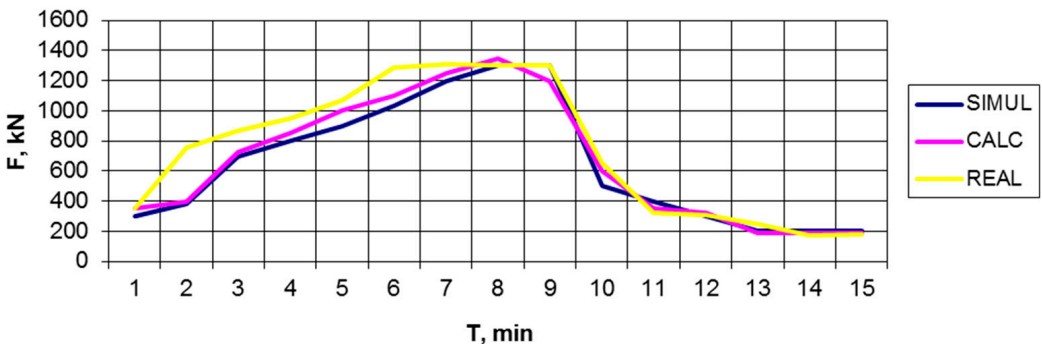

**Figure 21.** Tug towage line tension and pull/push forces during ship's unmooring, manoeuvring and departure operations, obtained by calculation (CALC) and simulation (SIMUL) during real operation with similar vessel under hydrometeorological conditions (REAL) (wind S—14 m/s).

Forces acting on the ship referred to in this article during unmooring, manoeuvring and departure from the bay, calculated by the methodology presented in this article, in the case of the southern wind of 14 m/s between 7 and 9 min of manoeuvring are presented in Table 2.

**Table 2.** Forces acting on ship (L = 238 m) during manoeuvring between 7 and 9 min.

| Forces | Inertia | Hull | Wing | Aero | Shallow | Propeller | Ruder | Tug |
|--------|---------|------|------|------|---------|-----------|-------|-----|
| *X*, kN | −5 | −10 | −5 | 100 | −100 | −50 | 1 | 70 |
| *Y*, kN | −10 | −120 | −10 | −750 | −480 | 20 | 100 | 1250 |

Based on the methodology presented in this article, the optimal bollard pull of tugs at the analysed port location for a vessel with a length of 238 m in the most unfavourable wind directions and speeds was calculated. The calculation results are presented in Figure 22.

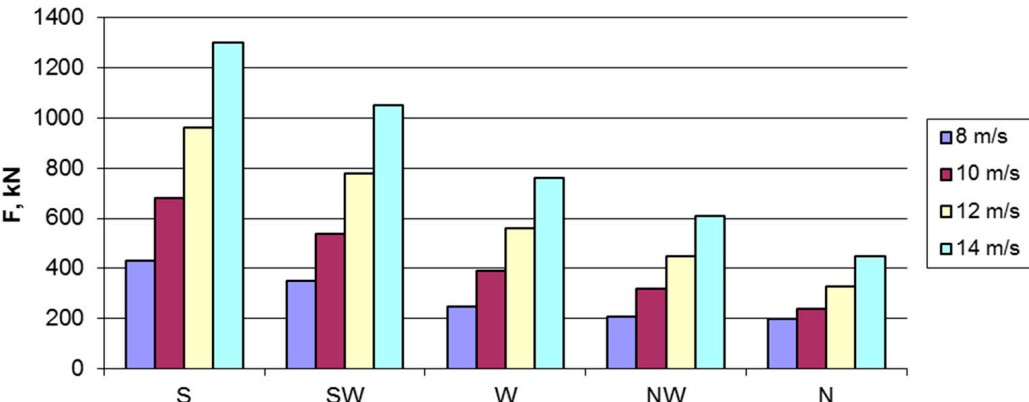

**Figure 22.** Optimal tugs bollard pull is required at the analysed port location for a vessel with a length of 238 m depending on the wind direction and speed.

During the manoeuvre to turn the ship in the bay against the south wind by using a push/pull method, both tugs (at the middle and stern parts of the ship) used 100% of their force. During similar real ship operation conditions, 3 tugs were used, and the maximum bollard pull used by all three tugs together was up to 1200–1300 kN (tugs used from 70 to 100% of their bollard pull and pushing during all unmooring and manoeuvring operations, especially during the ship's turn in the bay). It appears that the south direction wind is more complicated in these circumstances compared to the north or west direction wind.

This case study showed that the theoretical method of calculating the tug capacity and tug tensile strength (bollard pull) presented in the article fairly accurately (errors not exceeding 10 percent) describes the required optimal quantity of tugs and their tensile strength (bollard pull) under specific conditions and can be used for practical application in ports and shipping companies.

## 5. Discussion

The research results reveal that the qualifications of port pilots, ship masters and tug masters play a significant role in performing ship manoeuvres and, consequently, may well help to optimize the port tugs use [17,29,48].

A number of measurements conducted during the case study analysis can be discussed. Though this number was limited, it is still representative of the established research topic. The differences in operators' behaviour while performing manoeuvring operations were visible and proved that the qualification level of pilots and tug masters was different [3,7,26,45,49]. Therefore, it should be noted that the research results can be regarded as satisfactory and allowed us to answer the first research question, i.e., whether the optimization of the tug number and bollard pull at seaports is influenced by the qualifications of ship masters, tug masters and port pilots [3,11,12].

The experimental results showed that the capacity of the used tugs may be reduced by 10–15% or even more. Some literature sources mention, which was also proved by practical experience at ports, that the way the ship is manoeuvred may not only influence the capacity of the tugs being used but also decrease the volume of emissions from ships by up to 15–18% [56,57]. It should be also noted that the research presented in this article was conducted for the specific ship and defined sailing conditions limited to the port area, which also had an influence on the results. However, it was possible to answer the second research question and assess the capacity of the tugs that may be reduced during ship manoeuvring operations at the port area depending on the qualification of the persons in charge [3,37,48].

Moreover, research results may have managerial implications. Seaports, as well as shipping companies, may change their procedures and introduce strict conditions of skill verification during the employee hiring process and professional work, in pursuit of

reducing the volume of tug capacity and emissions at seaports. These activities may affect the development of companies' navigational safety and environmental policy in order to decrease the costs of the operation of ships, as well as the volume of emissions [3,17,29,48].

The achieved results also proved that the quality of maritime education is very important to ensure the necessary qualifications for ship operators. This justifies the need to increase the quality of professional education at universities, the theoretical knowledge and the number of practical hours on simulators among seafarers, which will enable an increase in their qualifications and attractiveness in the labour market [3,48,49].

## 6. Conclusions

Many ports have complicated navigation conditions, and port tugs must also assist the ships entering the port during mooring, unmooring and departure operations. The evaluation of the parameters of the ship's safety in advance is very important.

The methodology presented in this article could be successfully used for the optimisation of the use of port tugs under the complex ship mooring conditions whilst calculating an optimal request for tug bollard pull and tugs located along the ship's hull during complicated navigational conditions.

The forces and moments presented in this article and acting on ships and tugs at the port during the operations were checked using real ships under real conditions and showed a good correlation with the calculation results, since differences did not exceed 10%.

The analysis of the results obtained employing different calculation methods (classical and the one presented in this article) demonstrated the main advantage of the developed technique and indicated that it is possible to calculate the forces on the ship's bow and stern in advance, which can be used as a basis for ordering and locating optimal tugs before the ship enters the port and also during the mooring operations.

In-depth investigations and knowledge about forces and moments in complicated navigational conditions can increase navigational safety when the ships are entering the port, mooring and departing.

The study presented in this article aimed to develop a method to assess the possible optimization of the number of tugs and bollard pull considering the qualifications of the ship operators. This goal was achieved. The study results showed that the qualifications of the ship masters and port pilots can have an influence on the optimization of the use of port tugs.

The developed method enables the analysis of empirical data and may be introduced in practice. It shows the role of operators' education and training and justifies the need for regular improvement in staff qualifications. Moreover, the presented approach may be useful for seaports and shipping companies and may be implemented to assess the personnel's qualifications during the selection of staff responsible for ship steering.

**Author Contributions:** This paper was drafted and written by V.P., and all authors worked on the test and simulation results. D.P. and R.B. contributed to the query, determination and calculation of the simulation program. B.P. and M.S. provided guidance for the overall research ideas and plans. V.P. and M.J. provided guidance for the formulation and implementation of the test methods. All authors have read and agreed to the published version of the manuscript.

**Funding:** This research received funding from Klaipeda university doctoral studies.

**Institutional Review Board Statement:** Not applicable.

**Informed Consent Statement:** Not applicable.

**Conflicts of Interest:** The authors declare no conflict of interest. The funders had no role in the design of the study; in the collection, analyses, or interpretation of data; in the writing of the manuscript, or in the decision to publish the results.

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
