# Peer review of "The Influence of Port Tugs on Improving the Navigational Safety of the Port"

_jmse, doi:10.3390/jmse9030342_

Round 1

Reviewer 1 Report

The comments have been sufficiently addressed.

Author Response

Response to the Reviewer 2 (2021.03.09) of manuscript titled: “The influence of port tugs on improving the navigational safety of the port”

Dear Reviewer,

Thank you very much for the thorough and insightful review of our article and the suggestions to improve the quality of article.

We have the following comments to your suggestions:

Reviewer (1): Comments: 1.  English language and style are fine/minor spell check required.

Response: We agree with the remark and the revised text of the article. The English used in paper was reviewed and editing as well stylistic corrections took place. Additionally, we used the MDPI translation service and post-corrected the English language.

Thank you one more time for your involvement.

Authors

Reviewer 2 Report

Efforts to find meaning can be seen through many experimental data. Editing is not good, so new editing is requested. It is crude editing such as local language, spaces, and commas.
The description of the thesis argument should be logically changed. The purpose, method, and explanation of the data used and the meaning of the experimental results should be sufficiently conveyed.

Author Response

Response to the Reviewer 1 (2021.03.06) of manuscript titled: “The influence of port tugs on improving the navigational safety of the port”

Dear Reviewer,

Thank you very much for the thorough and insightful review of our article and the suggestions to improve the quality of article. The fragments that were changed in text are marked in yellow.

We have the following comments to your suggestions:

Reviewer (1): Comments: Efforts to find meaning can be seen through many experimental data. Editing is not good, so new editing is requested. It is crude editing such as local language, spaces, and commas.

Response: We agree with the remark and the revised text of the article. The English used in paper was reviewed and editing as well stylistic corrections took place. Additionally, we used the MDPI translation service and post-corrected the English language.

Reviewer (2): Comments: The description of the thesis argument should be logically changed. The purpose, method, and explanation of the data used and the meaning of the experimental results should be sufficiently conveyed.

Response: We agree with the comments. For a more logical presentation of the article, the introduction presents the purpose and meaning of data acquisition and use. The methodological part additionally provides a reference to the acquisition and use of data, i. three-step system (calculation, simulation and real ship experiment), and method of estimating the accuracy of the obtained data. The accuracy of the obtained data using the distribution method and reliability are indicated in the case analysis section.

Thank you one more time for your involvement.

Authors

Round 2

Reviewer 2 Report

It is judged that this study has been sufficiently revised at the request of the evaluator.

This manuscript is a resubmission of an earlier submission. The following is a list of the peer review reports and author responses from that submission.

Round 1

Reviewer 1 Report

Name of the paper is badly chosen. Tugboat cannot “influence” safety

Need thorough correction of the English language, grammar and words used. “Concrete” means building material and not “particular” as meant by the authors e.g

Ships can not impersonate human characteristics thus tugboats cannot “play the role”

When using expression “port tugs” it gives impression there have to be other types of tugs. Authors should give short explanation or tables defining tugboat and types of it

44 – if “port tugs” are used to assist vessels when maneuvering while in the port, how came they are improving navigational safety?

Figure 1 -  name of the table is confusing. Table must be explained in order to be used

65-68 it is not clear whether authors are making point that port of Klaipeda is having frequent port traffic or vessels are calling port of Klaipeda because fleet of port tugs are large with good particulars. Either way, nor the port or tugs can “attract” vessels

76-77 it is not possible that one tug boat can overreact forces generated by large vessel. Thus when pointing “tugs in the port shall be capable of generating forces greater than external forces of action on the vessels” must be stated how many tugs per respective vessels or, authors are trying to point out that combine bollard pull of all tugs in the fleet is enough for any vessel

Must define “external and internal” forces once the phrase is used in the paper

189 – must define if the reason for 80% of accidents and incidents is crew incompetence or poor preparations. Must give some valid argument justifying this “80% number”

389 – wall furniture? Obnoxious name for fenders

Winds are not “South wind” or “North wind”. Must use proper maritime terminology

510-512 there is not single research proving that qualifications of the masters on board ship and tugs as well as pilot’s qualification has any relevance to the results presented. It might be that competence of those mentioned could make some difference but again it is just a subjective opinion with out proper research and presented results of the same. Qualifications is something master and pilots must acquire and possess in order to be able to be in positions and it is certificate of competency, mandatory courses certificates etc etc

There were three situations simulated with same number of the tugs but different wind direction. Wind was always 14 m/s

Three different scenarios are not representative number for exhaustive research

There is no connection between chapter 3 “Theoretical basis for use of port tugs” and three scenarios on the case study

It is not clear what was authors aim. There are long chapter 3 where certain forces influencing vessels maneuverability are explained mathematically. Formulas are cited from numerous papers. Following chapter is describing three scenarios where only difference is direction of prevailing wind. In discussion authors are claiming that research results are revealing that qualifications of the captains on board vessel and tugs as well as pilots qualification is making significant difference.

This paper is poorly written and poorly structured. Discussion is not reflecting first three chapters, nor three simulations carried out.

The conclusion is that different usage of tugs in performed scenarios are because of prevailing winds and that is logical.

Reviewer 2 Report

The paper describes the application of a simulation model for escort operation, giving a worked example in the Klaipeda port. 

Hereafter some comments/suggestions:

  1. A better description should be given of the ships involved in the worked example.
  2. The quality of Figures 10,14, and 20 should be improved as actually are almost unreadable.
  3. The developed model is including the forces generated by the hull when acting with a drift angle? This is an issue more relevant for escort tugs rather than conventional tugs, however, a simulation model should include also these forces. The authors are encouraged to comment on this aspect.